# Enhanced Diffractive Circular Dichroism from Stereoscopic Plasmonic Molecule Array

**DOI:** 10.3390/nano13071175

**Published:** 2023-03-25

**Authors:** Liangliang Gu, Rong Shu, Xiangfeng Liu, Haifeng Hu, Qiwen Zhan

**Affiliations:** 1School of Optical-Electrical and Computer Engineering, University of Shanghai for Science and Technology, Shanghai 200093, China; 2Zhangjiang Laboratory, Shanghai 201204, China; 3Shanghai Key Lab of Modern Optical System, University of Shanghai for Science and Technology, Shanghai 200093, China; 4Key Laboratory of Space Active Opto-Electronics Technology, Shanghai Institute of Technical Physics, Chinese Academy of Sciences, Shanghai 200083, China

**Keywords:** Circular Dichroism, Chirality, plasmonics, Light Diffraction, Multipole Expansion

## Abstract

Artificial nanostructures with large optical chiral responses have been intensively investigated recently. In this work, we propose a diffractive circular dichroism enhancement technique using stereoscopic plasmonic molecule structures. According to the multipole expansion analysis, the z-component of the electric dipole becomes the dominant chiral scattering mechanism during the interaction between an individual plasmonic molecule and the plane wave at a grazing angle. For a periodical structure with the designed plasmonic molecule, large diffractive circular dichroism can be obtained, which can be associated with the Wood–Rayleigh anomaly. Such a diffractive circular dichroism enhancement is verified by the good agreement between numerical simulations and experimental results. The proposed approach can be potentially used to develop enhanced spectroscopy techniques to measure chiral information, which is very important for fundamental physical and chemical research and bio-sensing applications.

## 1. Introduction

Chirality refers to the geometrical feature, which describes an object that cannot be superimposed on its mirror image (enantiomer) [1]. Chiral objects are very common in nature. Many important biomolecules have chirality, such as amino acids, carbohydrates, and nucleic acids. For some drug molecules, the recognition and separation of the chirality are essential, because the enantiomers with the opposite chirality may be toxic to human bodies. Chiroptical spectroscopy techniques are used to evaluate the chirality of molecules [2]. The typical one is circular dichroism (CD), which can be calculated using CD = 2(W^+^ − W^−^)/(W^+^ + W^−^), where W+ and W− are the absorption/scattering powers under the illumination of left- and right-circularly polarized light (LCP and RCP). However, the CD signals from natural molecules and materials are extremely weak. In recent years, many artificial nanostructures have been designed to obtain strong optical chiral response [3,4,5,6,7,8] for broad applications, including light circular polarizers [9], chiral nanomotors [10], asymmetric synthesis [11], chiral nonlinear optics [12,13], chiral molecule sensing [14,15,16], and so on. To enhance the chiroptical response, various structures have been designed, such as nano helix, split-ring resonator, and chiral holes [17,18,19,20]. These plasmonic nano- particles have exotic optical properties, which stem from the collective oscillation of quasi- free electrons in the metal. Besides individual particles, groups of coupled metallic nano- particles termed “plasmonic molecules” with handed arrangements can also exhibit large chiral optical responses [3]. In addition, compared with the quasi-planar chiral structures fabricated using the one-step lithography method, stereoscopic structures can exhibit larger optical activity [21,22]. Multilayer structures have been stacked by achiral components to realize the three-dimensional plasmonic molecules to enhance the chiral optical response. J. Zhu et al. studied the strong light−matter interactions in gold nanorod- based chiral plexcitonic systems assembled on DNA origami [23]. Analogous to the Born–Kuhn model, a plasmonic molecule consisting of two orthogonal nanorods was designed [24,25]. The chiral response originates from the symmetric and antisymmetric modes excited by RCP and LCP, respectively. It can be found that many artificial chiral structures investigated in these previous works are periodical ones. Chiral signals are usually measured from the reflected and transmitted beams (i.e., zero-order beams), but high-order diffracted beams are rarely explored. V. K. Valev et al. demonstrated the chiral diffraction effect of meta-surface in the nonlinear regime [26]. The CD from the diffracted beam is orders of magnitude larger than that from the zero-order beam. C. Kuppe et al. measured the circular dichroism from the high-order diffracted beams of a chiral U-shaped nanostructure [27]. The structures in these two works are both quasi-planar with relatively small CDs. Therefore, the diffracted beam using a three-dimensional structure may be explored to further enhance the CD signal. Moreover, first- order diffraction has also been studied in numerous planar polarization gratings and the so-called Pancharatnam–Berry metasurfaces [28,29], which are the building blocks of contemporary meta-holograms and metalenses [30,31]. 

In this work, we designed a corner-stacked plasmonic molecule with a chiral geometry configuration. The multipole expansion method was employed to unveil the mechanism of the optical chiral response from an individual plasmonic molecule. Due to the coupling effect, the plasmonic molecule array exhibited different scattering properties from the individual one, especially for the case when the first diffraction order became evanescent, i.e., the condition of Wood–Rayleigh anomaly. For the plasmonic molecule array, the optical parameters that can be directly measured were the out-put powers carried by the reflected and diffracted beams. Both numerical and experimental results demonstrate that the CD from the diffracted beam was much larger than that from the zeroth- order reflected beam. The diffracted beam with enhanced CD signal can be further employed to probe the chirality of bio- and chemical molecules combined with periodical plasmonic structures.

## 2. Theoretical Design

The designed plasmonic molecule consists of a dielectric strip and two identical metallic strips. Its fabrication process is divided into four steps, as shown in Figure 1a–d. The dielectric strip with subwavelength scale should be first prepared. Then, the template of the resist layer is introduced to cover the dielectric strip. The two rectangular holes in the resist layer are formed using an electron beam lithography process. A thin metallic film is deposited on top of the resist layer. Defined by the template, two metallic strips are corner-stacked on the two ends of the dielectric one, so both of the metallic strips have stepped shapes. In the final step, the resist layer is removed using a lift-off process. The two metallic strips can be interchanged with a 180-degree rotation around the center point of the structure, so the plasmonic molecule has C2 symmetry. The geometrical parameters (width, length, and height) of the dielectric strip and metallic strip are represented by (*w_d_*, *l_d_*, and *h_d_*) and (*w_m_*, *l_m_*, and *h_m_*), as shown in Figure 1a,d, respectively.

The designed plasmonic molecule has geometrical chirality, so light scattering efficiencies should be different for the two incident beams with opposite handedness. To understand the mechanism of optical chiral response, the multipole expansion method is carried out when the illumination beams are LCP and RCP. The total scattering power can be decomposed as the summation of the contributions from multipole moments in the Cartesian coordinates [32,33].
(1)$$\begin{array}{l} W=\sum_{\alpha }\left(W_{\alpha }^{ED}+W_{\alpha }^{MD}\right)+\sum_{\alpha }\sum_{\beta }\left(W_{\alpha \beta }^{EQ}+W_{\alpha \beta }^{MQ}\right)+\cdots \\ \;\;\begin{array}{cc} & = \end{array}\sum_{\alpha }\left({\left|p_{\alpha }\right|}^{2}+{\left|\frac{m_{\alpha }}{c}\right|}^{2}\right)+\frac{1}{120}\sum_{\alpha }\sum_{\beta }\left({\left|kQ_{\alpha \beta }^{e}\right|}^{2}+{\left|\frac{kQ_{\alpha \beta }^{m}}{c}\right|}^{2}\right)+\cdots \end{array}$$
where *α*, *β* = *x*, *y*, *z*. *p_α_* and *m_α_* are the electric dipole (ED) and the magnetic dipole (MD) moments. $Q_{\alpha \beta }^{e}$ and $Q_{\alpha \beta }^{m}$ are the electric and magnetic quadrupole moments. *c* and *k* are the speed and wavevector of light in vacuum. For particles much smaller than the incident wavelength, the dominant terms are electric dipoles. The electric dipole moments can be determined using the integral of the current density J inside the plasmonic molecule as follows:(2)$$p_{\alpha }=-\frac{1}{i\omega }\left\{\int d^{3}rJ_{\alpha }j_{0}\left(kr\right)+\frac{k^{2}}{2}\int d^{3}r\left[3\left(r\cdot J\right)r_{\alpha }-r^{2}J_{\alpha }\right]\frac{j_{2}\left(kr\right)}{{\left(kr\right)}^{2}}\right\}$$
where ω is the angular frequency of light. *j_n_*(*x*) is the spherical Bessel function. The induced current density can be calculated through full-wave simulation based on the finite element method (FEM). In the numerical model, the material of the dielectric strip is silicon dioxide (*n_d_* = 1.46), and the geometrical parameters of the dielectric strip are *l_d_* = 200 nm, *w_d_* = 80 nm and *h_d_* = 50 nm. The two metallic strips are made from gold with *l_d_* = 140 nm, *w_d_* = 80 nm and *h_m_* = 50 nm, and the optical dispersion property of gold material is fitted using the Drude model [34]. The light scattering by an individual plasmonic molecule is considered, as shown in Figure 2a. The wavelength of incident light is 633 nm, and its wavevector is in the xz plane. *W*^+^ and *W*^−^ represent the total scattering powers under LCP and RCP plane waves, respectively. The simulation results of *W*^+^ and *W*^−^ versus incident angle are shown with the solid dash curves in Figure 2b,c. When the incident angle increases from 0° to 90°, the scattering power reduces dramatically. As expressed in Equation (1), the total scattering power can be decomposed via the radiation power by multipole moments. The intensity of the three ED components is calculated using Equation (2), as shown by the dash curves in Figure 2b,c. One can see that the scattering field is mainly radiated from the x-component of ED (i.e., *p_x_*) for the normally incident light. The intensity of *p_x_* is reduced with increasing incident angle. When the incident angle reaches 90°, the intensity of *p_x_* drops to zero. In Figure 2b, the intensity of *p_y_* is much smaller than *p_x_* and *p_z_*. When θ = 90°, the *p_z_* componentt, which originates from the geometrical features of the stereo-structure, is the main contributor to the scattering power *W*^+^. As shown in Figure 2c, under the illumination of RCP light, *p_z_* is strongly suppressed, and *p_y_* is excited in the plasmonic molecule. For grazing incident light, the dominant chiral scattering mechanism is the z-component of ED in the plasmonic molecule. In periodical structures, the grazing light can be generated under the condition of the Wood–Rayleigh anomaly.

We now consider the plasmonic molecule array with stereo-structure fabricated on the silicon substrate, as shown in Figure 3a. When LCP and RCP light beams are employed to probe the chirality of this structure, the reflected and diffracted powers can be detected directly. The directions of diffracted beams can be determined using the phase matching condition, i.e., *k_xn_* = *k_x_*_0_ − 2π*n*/*P*, where, *k_xn_* and *k_x_*_0_ are the x-components of the wavevectors of nth-order diffracted beam and incident beam. To simplify the light diffraction process, the proper period is selected to guarantee that only one diffraction order is generated. To calculate the diffraction efficiency of this structure, numerical simulation is carried out based on rigorous coupled wave analysis (RCWA). The normalized powers of reflected and diffracted beams are shown in Figure 3b–g. In this simulation, the influence of light frequency and incident angle are both investigated. The geometrical parameters of the plasmonic molecule remain unchanged, as shown in the model in Figure 2a. The periods of the chiral structure along the x-axis and y-axis are both 400 nm. The light frequency varies from 300 THz to 750 THz, with the corresponding wavelength range between 400 nm and 1000 nm. The incident beam is in the xz plane and the incident angle varies from 0° to 85°. In Figure 3b,c, the reflected power under the illuminations of LCP and RCP beams is plotted in the *k_x_*-frequency plane. The black solid line represents the light line, i.e., *ω* = *kc*. Because the incident beam propagates in free space, the region below the light line is not considered in this simulation. The circular dichroism from the reflected power is shown in Figure 3d. An enhanced CD is observed with the value of 0.16 associated with the Wood–Rayleigh anomaly, which is indicated by the dash ed line. The bandwidth of the CD resonance peak is 20 nm at the incident angle of 35.6°. A similar effect has also recently been reported in the structure of periodical dielectric dimers [19]. In the region above the dashed line, first- order diffraction is involved in the interaction between light and the periodical structure. The diffraction spectra for LCP and RCP incident beams are demonstrated in Figure 3e,f, respectively. There is a peak of diffracted power near the light frequency of 500 THz, which indicates a strong interaction between light and the plasmonic cell. In Figure 3g, a remarkable enhancement is observed in the CD from the diffracted beam, compared with the reflected beam. This can be partially attributed to the influence of the substrate. The reflected beam is composed of the beam directly reflected by the substrate and the scattered beam in the same direction. The planar substrate has no chirality, so the reflected power contributed by the substrate remains unchanged when the incident beam is switched between LCP and RCP beams. The nonchiral reflected power by the substrate can reduce the CD signal. However, the diffracted beam is only produced by the scattering light from the plasmonic cells, and the substrate reflection can be eliminated in the CD measurement. The CD enhancement exists near the Wood–Rayleigh anomaly condition marked by the black dashed line. There are several peaks of CD values along the black dashed line in Figure 3g. Considering the diffraction efficiency of the first-order beam, the CD peak with the frequency of 474 THz (i.e., λ = 633 nm) is selected for further experimental study.

## 3. Experimental Implementation

Following the steps in Figure 1, the dielectric strip in the designed stereoscopic structure was fabricated first. The thin film of silicon dioxide with a thickness of 50 nm was deposited on the silicon substrate using plasma-enhanced chemical vapor deposition (CC-200CZ, ULVAC). Then, electron beam lithography (ELS125, Elionix) and reaction ion etching (R150S, Haasrode) were carried out to define the shape of the silicon dioxide strip array with a period of 400 nm. The resisted layer with a thickness of 200 nm was then introduced on the wafer via spin coating. EBL was employed again to generate the template of metallic strips in the resisted layer. The patterns of the template and the dielectric strip array should be aligned to make sure the metallic strips can be corner-stacked on the dielectric strip, as shown in Figure 1. After that, a Ti/Au film was deposited on the wafer via electron beam evaporation. The Ti film was the adhesion layer with a thickness of 10 nm. The thickness of the Au layer was 30 nm. The resist layer was finally removed through a lift-off process. The area of the fabricated structure was 50 μm × 50 μm. All the geometry parameters of the plasmonic molecule were confirmed via scanning electron microscopy (Helios G4-UX, Thermo Fisher Scientific, Waltham, MA USA). The image of the plasmonic molecule array from the top view is shown in the inset of Figure 4a. An experiment was set up as shown in Figure 4a to measure the CD signal. A continuous wave (CW) semiconductor laser with an output wavelength of 633 nm was employed as the light source, focused by a lens with f = 50 mm. The diameter of the focused spot of the laser beam at the position of the sample was 150 μm, measured with a CCD camera. The laser beam then passed through a polarizer and a liquid crystal variable retarder (LCVR) (LCC1411-A, Thorlabs). These two devices’ optical axes are indicated by the white dashed line in Figure 4a, and the angle between them was 45°. A polarizer was used to force the light beam polarized along the vertical direction. When the linear polarized beam passed through LCVR, the output polarization state can be modified depending on the phase retardation *ϕ* of LCVR, as shown in Figure 4b. The value of *ϕ* can be tuned using the electric voltage applied on LCVR. When the *ϕ* was 90° and 270°, the LCP and RCP beams were generated. The sample was fixed on a rotation stage, which could adjust the incident angle of the laser beam. A photodetector (PDA100A2, Thorlabs) was used to measure the power of the diffraction beam.

In our experiments, the diffracted power was measured when the phase retardation of LCVR was tuned from 0° to 300°, which is limited by the modulation range of the device. As a comparison, the diffracted power under differently polarized light was also calculated under the same illumination condition. A position misalignment was observed between the dielectric strip and metallic strips, which was caused by the small misalignment between the template pattern and the dielectric strip in the second EBL process. To consider the influence of this fabrication error, the dielectric strip also shifted in the numerical model. The position shifted along the x and y axes with Δx = −10 nm and Δy = 10 nm. In Figure 5a,b, the simulated and experimental results for the first-order diffracted beam are compared. The incident beam was in the xz plane, and the incident angle wais θ = 40°, 50°, 60°, and 70°. The CD value can be extracted from the curves in Figure 5a,b. The CD from the diffracted beam was reduced when the incident angle increased, as shown in Figure 5c. The diffracted power for the incident angle larger than 70° was not measured, because the incident light was blocked by the clamp to hold the sample in the setup. The incident beam within the yz plane is also considered in Figure 5d–f. The experimental results clearly demonstrate that the large CD can be produced with the diffracted beam from the designed plasmonic molecule array. The maximum CD of 0.556 was achieved in this work. There were some deviations in the numerical results when comparing them with the measured values. This can be explained by the non-perfect geometric structures, and the discrepancies between the true and the textbook values of the refractive index for different materials.

## 4. Conclusions

In conclusion, we demonstrated an enhancement in CD by measuring the diffracted power from the designed stereoscopic plasmonic molecule array. The multipole expansion analysis was carried out to study the mechanism of the enhanced chiral optical scattering. When illuminated at large oblique incident angles, the chiral scattering field was mainly attributed to the z-component of ED in the stereoscopic plasmonic molecule. To enhance the interaction between the grazing light and the chiral structure, the optical response of the plasmonic molecule array was investigated through both numerical analysis and experiments. It was found that the CD measured from diffracted light can be highly enhanced, compared with the results from zeroth- order reflected light, especially when the condition of the Wood–Rayleigh anomaly was fulfilled. Compared with the previous works about the diffracted beams from chiral gratings [27], we employed the stereoscopic structure to further enhance the chiral response. The maximum CD measured in this work was 0.556. This value is much higher than the 0.1~0.2 values reported in [27] and can be further improved by optimizing the design of the plasmonic chiral molecule. The high chiroptical performance may be utilized to detect the polarization states of light beams and recognize the chirality of bio-molecules. Moreover, considering the broadband CD enhancement, the designed structure can be used as the dispersive element for the development of a high-performance CD spectrometer with a compact size.

## Figures and Tables

**Figure 1 nanomaterials-13-01175-f001:**
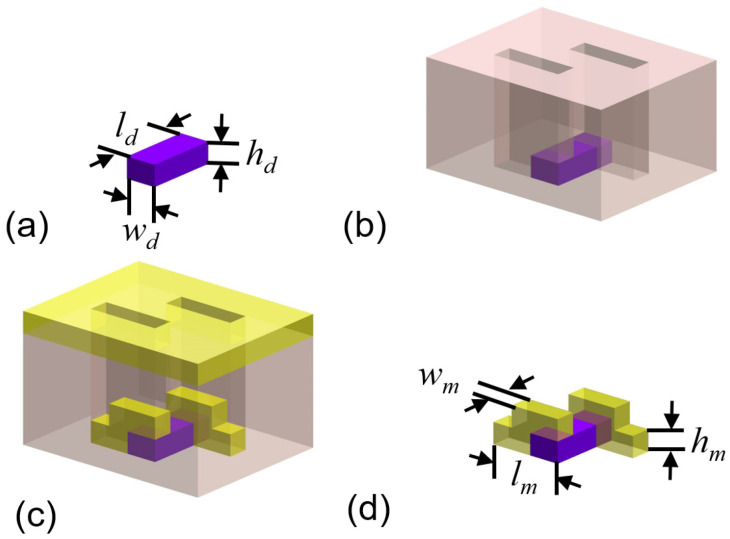
The schematic of the fabrication steps for the plasmonic molecule with stereo-structure. (**a**) The dielectric strip with the geometrical parameters of *w_d_*, *l_d_* and *h_d_*. (**b**) the resist layer with two rectangular holes is the template for the following step. (**c**) metallic film deposition. (**d**) the final plasmonic stereo-molecule after removing the resisted layer. The geometrical parameters of the metallic strip are *w_m_*, *l_m_* and *h_m_*.

**Figure 2 nanomaterials-13-01175-f002:**
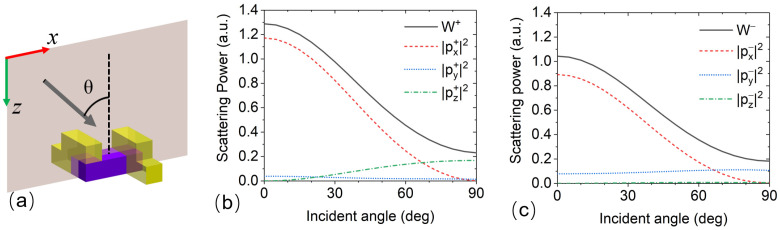
(**a**) The schematic of a single plasmonic molecule illuminated using LCP and RCP light. The wavevector is in the xz plane, and the angle between incident direction and z-axis is θ. (**b**,**c**) the total scattering power (solid curves) and the intensities of the three components of electric dipole (dash curves), when the incident light is LCP and RCP, respectively.

**Figure 3 nanomaterials-13-01175-f003:**
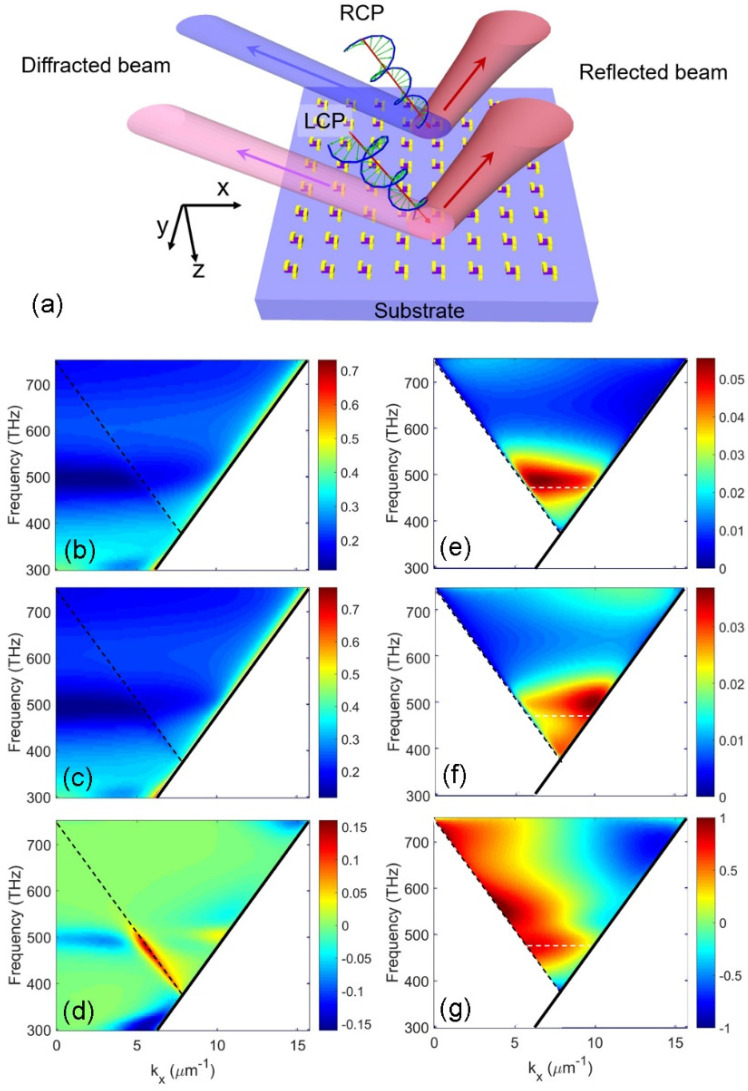
(**a**) The chiral plasmonic molecule array on the substrate. The reflected beams (indicates by red arrows) and the diffracted beams (indicates by blue and pink arrows) can be excited by the illuminations of LCP and RCP beams (in the xz plane). The reflection efficiencies of the chiral stereo- structure array by LCP beam in (**b**) and RCP beam in (**c**). The CD from the reflected beam in (**d**), which is calculated using the values in (**b**,**c**). The diffraction efficiencies with the LCP beam in (**e**) and RCP beam in (**f**), and the CD from the diffracted beam in (**g**) are shown. The black line represents a light line in free space. The black dash ed line represents the position of the Wood–Rayleigh anomaly. The white dash line in (**g**) indicates the frequency position with 474 THz (λ = 633 nm).

**Figure 4 nanomaterials-13-01175-f004:**
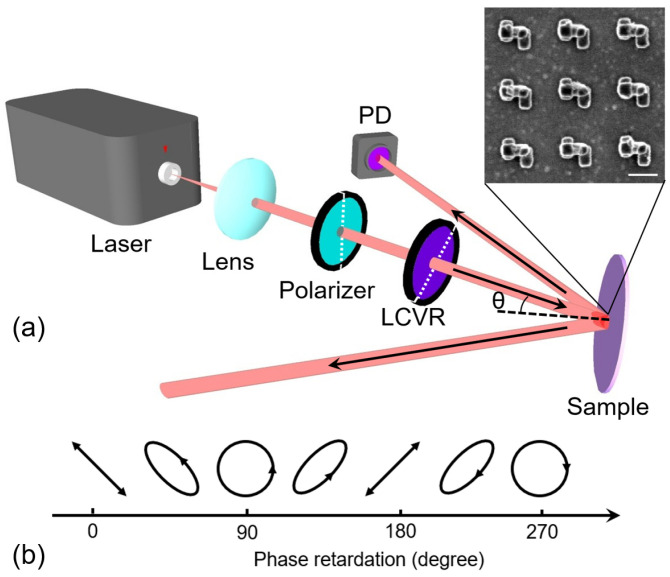
(**a**) The experimental setup to measure the CD from diffracted beams. The incident angle is θ. The power of the diffracted beam was collected using a photodetector (PD). The inset demonstrates the SEM image of the chiral stereo-structure, and the scale bar is 200 nm; (**b**) the evolution of the polarization states of incident beams versus the phase retardation introduced using an LCVR device.

**Figure 5 nanomaterials-13-01175-f005:**
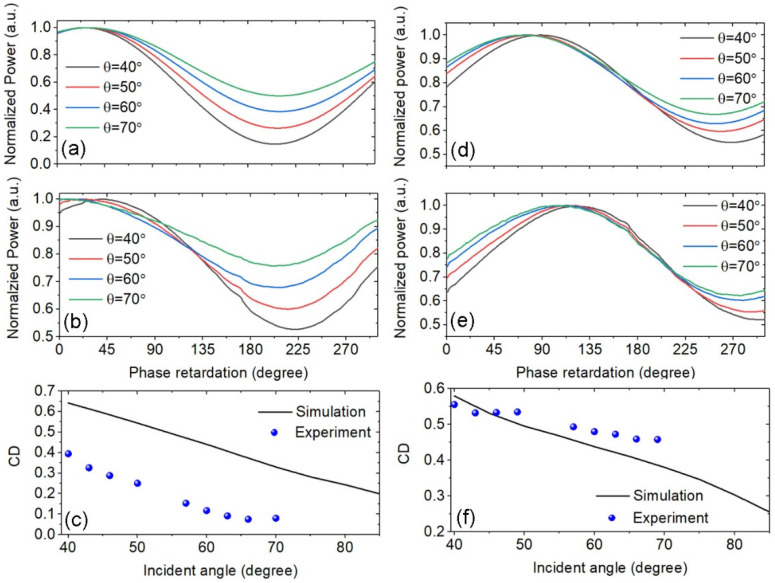
The numerical results for diffraction power versus phase retardation of LCVR in (**a**,**b**); the measured diffraction power in (**c**,**d**); the angular dependence of CD values in (**e**,**f**); (**a**–**c**) the incident beam is in the xz plane; (**d**–**f**) the incident beam in the yz plane.

## Data Availability

The data that support the findings of this study are available from the corresponding author upon reasonable request.

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
