# Peer review of "Enhanced Diffractive Circular Dichroism from Stereoscopic Plasmonic Molecule Array"

_nanomaterials, 2023, doi:10.3390/nano13071175_

Round 1
Reviewer 1 Report
Gu and colleagues reported a diffractive circular dichroism enhancement technique using stereoscopic plasmonic molecule structures. The authors studied the enhanced diffractive circular dichroism experimentally and theoretically. This manuscript looks good and I would recommend minor revision after a few corrections listed below.
Comments and questions:
1. In the Introduction section, there are few previous publications on this subject. I recommend that the authors add more relevant publications discussing CD values form diffractive.
2. Is the stereo-structure in Figure 1 new? If not, more previous publication is needed.
3. In Figure 2(a), is the angle theta in the z-direction orthe x-direction?
4. The scheme In Figure 3(a) is confusing. Does the RCP beam irradiate within the yz plane or the xz plane?
Reviewer 2 Report
The main achievement reported in this paper is the development of new nanofabrication technique allowing creating peculiar 3D shaped objects consisting of superimposed bars of different materials (silicon dioxide and gold). The idea is fresh and nicely expands the palette of complex elements of photonic metasurfaces feasible for fabrication. The great demand for such techniques is motivated by numerous theoretical works promising very broad application prospects of metasurfaces consisting of non-planar meta-atoms. In this context I find publication of the paper in Nanomaterials MDPI justified.
The reported optical properties are, in contrast, far from revolutionary. Very different planar metasurfaces are capable of selective diffraction of left and right circular polarizations. The authors compare their supposedly very strong circular dichroism (CD) of the diffraction to the achievements of paper [27]. The paper [27], however, reports strong CD in high (e.g. 4th) diffraction orders. CD of the 1st diffraction order is a much more trivial phenomenon and can be produced by numerous planar polarization gratings and so-called Pancharatnam–Berry metasurfaces, see e.g. [R1,R2], which are the building blocks of contemporary metaholograms and metalenses, see e.g. [R3, R4] .
It is a pity, of course, that the authors could not invent more fascinating application of their fabrication technique. I recommend at least to put the current modest achievements in the proper context.
[R1] Bomzon, Z., Biener, G., Kleiner, V. & Hasman, E. Space-variant Pancharatnam–Berry phase optical elements with computer-generated subwavelength gratings. Opt. Lett. 27, 1141 (2002).
[R2] 1. Yu, N. & Capasso, F. Flat optics with designer metasurfaces. Nature Mater 13, 139–150 (2014).
[R3] Ke, Y. et al. Optical integration of Pancharatnam-Berry phase lens and dynamical phase lens. Appl. Phys. Lett. 108, 101102 (2016).
[R4] Choudhury, S. et al. Pancharatnam–Berry Phase Manipulating Metasurface for Visible Color Hologram Based on Low Loss Silver Thin Film. Advanced Optical Materials 5, 1700196 (2017).
